# The evaluation of a web-based tool for measuring the uncorrected visual acuity and refractive error in keratoconus eyes: A method comparison study

**Marc B. Muijzer**[1,2☯]*, **Janneau L. J. Claessens**[1☯], **Francesco Cassano**[2], **Daniel A. Godefrooij**[1], **Yves F. D. M. Prevoo**[2], **Robert P. L. Wisse**[1,2]*

**1** Utrecht Cornea Research Group, Ophthalmology Department, University Medical Center Utrecht, Utrecht, Netherlands, **2** Easee BV, Amsterdam, Netherlands

☯ These authors contributed equally to this work.
* m.b.muijzer@umcutrecht.nl (MBM); r.p.l.wisse@umcutrecht.nl (RPLW)

**Data Availability Statement:** The data are available from the Open Science Framework database (DOI:

## Abstract

### Purpose

To evaluate the outcome of a web-based digital assessment of visual acuity and refractive error, compared to a conventional supervised assessment, in keratoconus patients with complex refractive errors.

### Material and methods

Keratoconus patients, aged 18 to 40, with a refractive error between -6 and +4 diopters were considered eligible. An uncorrected visual acuity and an assessment of refractive error was taken web-based (index test) and by manifest refraction (reference test) by an optometrist. Corrected visual acuity was assessed with the prescription derived from both the web-based tool and the manifest refraction. Non-inferiority was defined as the 95% limits-of-agreement (95%LoA) of the differences in spherical equivalent between the index and reference test not exceeding +/- 0.5 diopters. Agreement was assessed by a Bland-Altman analyses.

### Results

A total of 100 eyes of 50 patients were examined. The overall mean difference of the uncorrected visual acuity measured -0.01 LogMAR (95%LoA:-0.63–0.60). The variability of the differences decreased in the better uncorrected visual acuity subgroup (95%LoA:-0.25–0.55). The overall mean difference in spherical equivalent between the index and reference test exceeded the non-inferiority margin: -0.58D (95%LoA:-4.49–3.33, P = 0.008). The mean differences for myopic and hyperopic subjects were 0.09 diopters (P = 0.675) and -2.06 diopters (P<0.001), respectively. The corrected visual acuities attained with the web-based derived prescription underachieved significantly (0.22±0.32 logMAR vs. -0.01±0.13 LogMAR, P <0.001).

0.17605/OSF.IO/R2FPQ, link: https://doi.org/10.
17605/OSF.IO/R2FPQ).

**Funding:** This investigator initiated study was
sponsored by Easee BV. The funder provided
support in the form of salaries for author [FC], but
did not have any additional role in the data analysis,
decision to publish, or preparation of the
manuscript. The specific roles of these authors are
articulated in the 'author contributions' section.

**Competing interests:** MM is a consultant for Easee
BV, FC is an employee of Easee BV, YP is the CEO/
founder and shareholder of Easee BV, RW is a
consultant and shareholder of Easee BV. The
competing interest of the authors do not alter our
adherence to PLOS ONE policies on sharing data
and materials.

**Abbreviations:** AK, Amsler Krumeich; CDVA,
Corrected distance visual acuity; CI, confidence
interval; D, diopter; ETDRS, Early Treatment
Diabetic Retinopathy Study; LoA, limits of
agreement; LogMAR, logarithm of the minimum
angle of resolution; GEE, generalized estimates
equation; UDVA, uncorrected distance visual
acuity; ICC, intraclass correlation coefficient.

## Conclusions

Regarding visual acuity, the web-based tool shows promising results for remotely assessing
visual acuity in keratoconus patients, particularly for subjects within a better visual acuity
range. This could provide physicians with a quantifiable outcome to enhance teleconsulta-
tions, especially relevant when access to health care is limited. Regarding the assessment
of the refractive error, the web-based tool was found to be inferior to the manifest refraction
in keratoconus patients. This study underlines the importance of validating digital tools and
could serve to increase overall safety of the web-based assessments by better identification
of outlier cases.

## Introduction

Globally, an estimated 1 billion people have a visual impairment that can be prevented or is
undetected and this number is expected to rise [1, 2]. Main causes of visual impairment are
refractive errors, cataract or chronic ophthalmic conditions (e.g. macular degeneration). In an
aging population, the demand for eye care is increasing rapidly, which imposes a challenge for
providers of eye care in both developed and less-developed countries [1, 3]. The acutely
reduced access to healthcare during the COVID-19 outbreak underlined the need for a para-
digm shift in the delivery of eye care [4].

To become less reliant on hospital facilities and trained professionals for monitoring eye
conditions, there is a growing interest in telemedicine and digital tools. In particular, promis-
ing advances have been made for automatic assessment of retinal images [5]. The refractive
error and visual acuity are considered important clinical parameters for diagnosing and moni-
toring eye conditions and as such various tools to remotely assess refractive errors and visual
acuity are developed and clinically validated [6, 7].

Currently, the manifest refraction–consisting of the measurement of the refractive error
and the visual acuity–by a trained professional is considered the gold standard [8]. Automated
assessment of the refractive error using an autorefractor is considered non-inferior to the man-
ifest refraction in healthy eyes [9, 10]. However, the reliability of the autorefractor decreases in
eyes suffering from complex refractive errors (a hallmark sign of keratoconus) and it does not
measure visual acuity [11]. Moreover, both the manifest and automated techniques require
(expensive) medical equipment and/or qualified personnel and makes them unsuitable for
home monitoring or use in less-developed countries.

Recently the authors published the outcomes of the"manifest versus online refraction evalu-
ation"-trial, which reports the validation of a web-based refractive assessment in healthy adults
[6]. The tool was found non-inferior to a manifest refraction performed by a trained optome-
trist and is accessible via https://easee.online. Notably, the tool does not require any specialized
equipment–it only requires a mobile phone and a computer screen–and thus can be performed
in most home environments. However, the authors acknowledge that the results within a
healthy population are not necessarily representative for individuals with a suboptimal visual
performance. Therefore the design of the MORE-trial included a cohort of keratoconus
patients to evaluate the outcome of the digital refraction tool in a population with an eye con-
dition. Keratoconus was chosen as these patients are often still able to achieve an acceptable
visual acuity with a proper prescription, despite their complex refractive errors, including
irregular astigmatism [11, 12]. Keratoconus is an uncommon condition typically diagnosed in
adolescents, where gradual thinning of the cornea leads to a progressive ectasia and subsequent

irregular astigmatism [13, 14]. With its insidious onset, many early patients are unaware of this diagnosis, and keratoconus patients therefore pose a challenge for any assessment of refractive error [11].

Here, we present the results of the keratoconus cohort of the MORE trial, a validation study of a web-based tool for the assessment of visual acuity and refractive error.

## Material and methods

### Study design and recruitment

Data were prospectively collected in the single-center method comparison 'manifest versus online refraction evaluation (MORE)'-trial, performed at the University Medical Center Utrecht in Utrecht, the Netherlands. This study consisted of two subgroups; healthy individuals (n = 100) and keratoconus patients (n = 50). The outcomes of the web-based tool in healthy individuals is reported elsewhere [6]. The current manuscript pertains to the keratoconus patients of the MORE trial. Participants enrolled into the study were patients who visited the keratoconus clinic at the University Medical Center Utrecht in Utrecht. Inclusion criteria were an established diagnosis of keratoconus (diagnosed by an ophthalmologist based on clinical signs and Scheimpflug corneal tomography and graded using the Amsler-Krumeich classification), a clear central cornea, and an age between 18 to 40 years [13]. Subjects were excluded if their manifest refractive error, converted to spherical equivalent, was worse than -6 diopter (D; for myopia) or +4 D (for hyperopia), or with co-existing visual acuity limiting conditions. The boundaries regarding both age and refractive error are determined by the technical & regulatory limits of the web-based tool. Furthermore, we excluded subjects who had undergone corneal crosslinking 6 months prior to study participation to diminish effects of corneal haze, who had diabetes, who were pregnant or lactated, or who were unable to perform the web-based refraction assessment. Consecutively presenting patients that matched our criteria were invited to participate in the study (Fig 1).

All procedures were performed in accordance with the Declaration of Helsinki, local and national laws regarding research (i.e. the Act on Scientific Research Involving Humans), European directives with respect to privacy (General Data Protection Regulation 2016/679) and medical devices (Medical Device Regulation 2017/745), and the 2015 Standards for Reporting Diagnostic Accuracy Studies [15]. The study protocol was approved by our institution's Ethics Review Board (Medical Ethical Review Committee Utrecht; number: 17–524), and it was registered at clinicaltrials.gov (number: NCT03313921) and the CCMO (number: NL61478.041.17). All participants provided written informed consent and were enrolled in the study between January 31, 2018 and June 23, 2019.

According to the MORE trial protocol, all subjects underwent three consecutive tests designed to determine the refractive state of both eyes in the following order and the subject was blinded for the outcome of all tests. First, the refractive error was measured using autorefraction (Topcon RM 8800, Topcon Corporation, Japan) and corneal imaging was performed using Scheimpflug tomography (Pencatam HR, Oculus GmbH, Germany). Second, an optometrist performed the reference test (manifest subjective refraction). Finally, the subject performed the index test using the digital refraction tool. The digital refraction tool is web-based and was a custom version of the commercially available Easee refractive assessment tool, specifically built for this clinical trial and is identical to the second-generation algorithm we have described previously [6]. In short, a smartphone functions as a remote control by which the user submits input from a distance of 3 meters to a computer screen that displays the Web-based assessment. The user is presented a sequence of optotypes and astigmatism dials. Any visual acuity below 1.0 (worse than 20/20) is considered to be caused by a refractive error. The web-based refraction assessment is classified as a *Conformité Européenne* class 1m medical

STARD flow diagram

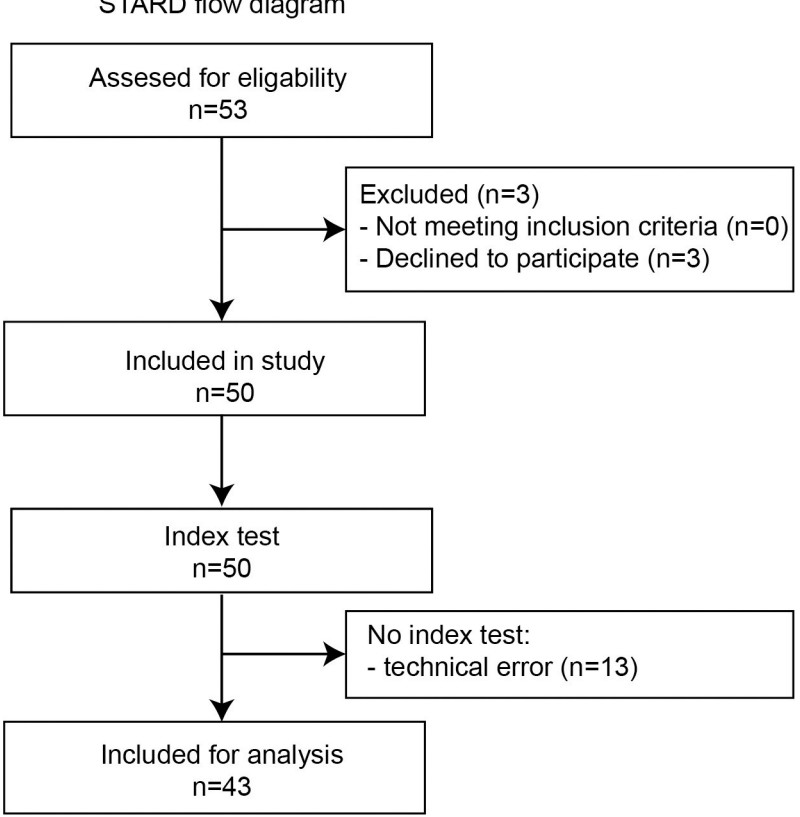

**Fig 1. STARD flow diagram illustrating participant flow of the keratoconus population of the MORE-trial.** All included participants underwent the web-based (index test) and manifest assessments (reference test) of visual acuity and refractive error.

device, which is in accordance with European Union Medical Device Regulation 2017/745, and the software is classified as class A, which is in accordance with International Electro technical Commission standard 62304:2014.

The uncorrected distance visual acuity (UDVA) was recorded using an Early Treatment Diabetic Retinopathy Study (ETDRS) visual acuity wallchart and the web-based visual acuity test. Corrected distance visual acuity (CDVA) was measured using a correction based on the results of the manifest and web-based refraction assessment outcome. Visual acuity was tested in accordance with ISO 8596, with regard to optotypes and room illumination [16]. The projected optotypes were randomized to mitigate any possible test-retest effect. The study protocol did not cover the assessment of CDVA with the autorefraction result, as previous research has shown a low reliability of autorefraction measurement in keratoconus eyes [11].

The following data were recorded for each participant/eye: age, gender, laterality, (ophthalmic) medical history, Amsler-Krumeich stage (AK1: mild, AK2: moderate, AK3: severe), mean & maximum keratometry, previous prescription (if known), use of spectacles or contact lenses, UDVA, CDVA, and refractive outcome, including spherical and cylindrical power (in D) and axis (in degrees).

## Statistical analysis

The primary study outcome was refractive error as measured using the web-based refractive assessment and compared with the subjective manifest refraction and autorefraction. The

signation of the refractive error (+/-), spherical power, cylindrical power were converted into a spherical equivalent. In concordance with the MORE trial protocol, the agreement of the spherical equivalent of the various methods was compared using a two-way mixed effect intra-class correlation coefficient (ICC) of a single measurement. In addition, the difference between the measurements of the web-based and manifest assessment were compared using a Fourier analysis. Specifically, we analyzed the signation of the refractive error (+/-), spherical power, cylindrical power, and axis, which were converted into power vectors [10, 17, 18]. Subsequently, the difference between the power vectors of the various methods was calculated as a residual vector (i.e. a vector of the difference) [11]. The power vectors are non-linear in nature, which precludes statistical analysis of differences between power vectors. Secondary study outcomes included the UDVA and CDVA, the latter measured using the outcome of both the manifest and web-based refractive assessment. A post-hoc subgroup analysis was performed for high and low uncorrected visual acuity ranges (high: $\leq$0.5 logMAR, low >0.5 LogMAR; based on ETDRS chart outcomes), with a cut-off as defined by the World Health Organization [19].

The data were assessed for normality of the distribution. Differences with a *P* value <0.05 were considered statistically significant. The outcomes were stratified for myopia, hyperopia, and keratoconus stages. Groups were compared using the two-tailed paired Student's *t* test. Non-inferiority was defined as the 95% limits of agreement of the difference in spherical equivalent between the web-based refractive assessment and the manifest refraction not exceeding ±0.5D, assessed using a Bland-Altman analysis [20]. In addition, a multivariable generalized estimates equation (GEE) analysis of the difference between the power vectors, was used to correct for bilaterality (both eyes of the same patient included), keratoconus severity, age, and sex. For keratoconus severity, Amsler-Krumeich stages 2 and 3 were combined, due to the low number of cases. Missing cases were not included in the analysis nor imputed.

The power calculation of the MORE-trial addressed the outcome in healthy subjects (n = 100), not in keratoconus subjects (n = 50). For methodological clarity we performed a post-hoc sample size calculation to determine if there was sufficient power to assess the non-inferiority limit of ± 0.5D. Here we assumed no difference between the web-based and manifest refraction assessment, an accepted differences of 0.5 D and a SD of 1.64 D. Using an alpha of 0.05 and a power of 0.90, and based on a one-sided one-sample t-test, 93 eyes are required. The actual study size of the keratoconus population (n = 100 eyes) was therefore considered sufficient to reject a null hypothesis. Data were analyzed using SPSS version 25.0 (IBM, Armonk, New York, USA) and R statistical software version 4.0.3 (CRAN, Vienna, Austria). For the Bland-Altman analysis the "BlandAltmanLeh" (version 0.3.1) package was used.

## Results

A total of 100 eyes from 50 keratoconus patients were included in the study, no subjects were excluded. The clinical characteristics of the study population are summarized in Table 1 and all relevant variables are stratified for Amsler-Krumeich keratoconus stage [13]. The majority of the participants was male (78%, n = 39), and used visual aids to correct their refractive error (78%, n = 39). Both results were expected; keratoconus is more prevalent in males and the hallmark cone shape in keratoconus induces myopia and (irregular) astigmatism [13]. A total of 23 underwent corneal crosslinking (n = 37 eyes) with >6 months follow-up, all resolved without sequelae. A total of 9 subjects reported ocular complaints at the time of the measurement; all 9 subjects reported blurred vision. No adverse events or complications were recorded during the trial. The refractive error and visual acuity data was missing in 13 subjects (11 myopic and 2 hyperopic) because of technical errors in the web-based refractive assessment.

**Table 1. Clinical characteristics of the study population (100 eyes of 50 patients).**

| Age (years), mean ± SD | 25.6 ± 5.4 |
|---|---|
| Sex (male), n (%) | 39 (78) |
| Current use of visual aids, n (%) | 38 (78) |
| Spectacles, n (%) | 28 (56) |
| Contact lenses, n (%) | 17 (34) |
| Ocular complaints, n (%) | 9 (18) |
| Medication use, n (%) | 10 (20) |
| Previous corneal crosslinking (CXL) treatment, n (%) | 24 (48) |

**Distribution of refractive errors and keratoconus severity classification[a]**

|  | Total (n = 100) | Mild keratoconus | Moderate keratoconus | Severe keratoconus |
|---|---|---|---|---|
|  |  | AK stage 1 (n = 84) | AK stage 2 (n = 13) | AK stage 3 (n = 3) |
| Mean keratometry, mean ± SD | 45.66 ±2.90 | 45.27 ±2.79 | 47.46 ±1.75 | 48.63 ±5.37 |
| Maximum keratometry, mean ± SD | 52.55 ±6.39 | 52.08 ±6.30 | 54.50 ±4.48 | 57.33 ±13.68 |
| Refractive error[1] | 100 | 84 | 13 | 3 |
| Hyperopia, n | 29 | 23 | 5 | 1 |
| Emmetropia, n | 5 | 5 | 0 | 0 |
| Mild myopia, n | 44 | 40 | 4 | 0 |
| Severe myopia | 22 | 16 | 4 | 2 |

Abbreviations: AK; Amsler-Krumeich stage, SD; standard deviation.

[a] Mild myopia was defined as refractive error of –3 diopter or less; severe myopia was defined as refractive error worse than –3 diopter. Refractive error was determined on the basis of the spherical equivalent of the manifest refraction value, (reference test), and is reported for both eyes separately. The distribution of refractive errors was not significantly different between the classifications ($P = 0.30$).

## Web-based visual acuity testing

UDVA was measured using the digital index test and with an ETDRS visual acuity wall chart as reference. Mean UDVA measured digitally was logMAR 0.57±0.39 (Snellen 0.38±0.33), and with the ETDRS wall chart logMAR 0.58±0.52 (Snellen 0.46±0.40). The mean difference between the measurements was considered small and non-significant (-0.01 LogMAR, $P = 0.76$). There was a considerable distribution in measurement differences (95% LoA: -0.63–0.60). In the better visual acuity subgroup ($\leq$0.5 LogMAR) this variability decreased (mean difference: 0.15 LogMAR, 95% LoA: -0.25–0.55). In the low visual acuity subgroup the mean difference between both measurements was -0.20 LogMAR (95% LoA: -0.82–0.41). Fig 2 depicts the correlation of the web-based visual acuity assessment compared with the ETDRS measurement in a Bland-Altman plot. The visual acuity assessments do not agree equally through the range of measurements. As can be observed in the high visual acuity subgroup, the majority of the measures show little difference and, importantly, the mean difference is mostly attributed to outliers. In this visual acuity range, the web-based tool slightly underestimates visual acuity scores. In the lower visual acuity range, the measurements appear to be much more variable.

## Web-based refractive error assessment

**Intraclass correlation coefficients.** The concordance of the refractive error among the refraction assessments was assessed using the ICC of the spherical equivalent. The overall ICC of all 3 assessments was 0.32 (95% CI 0.19–0.46). The ICC for the manifest refraction and web-based refractive assessment was overall 0.36 (95% CI 0.22–0.53) and for the mild keratoconus subgroup 0.48 (95% CI 0.29–0.64). All eyes were included in the ICC because of the asymmetrical manifestation of keratoconus. ICC calculations for either all right or left eyes separately did not lead to new insights.

**Fig 2. A Bland-Altman plot displaying the differences in logarithmic minimum angle of resolution (LogMAR) between the web-based uncorrected distance visual acuity assessment (index test) and the ETDRS uncorrected distance visual acuity measurement (reference test).** The differences between the reference test and index test shown on the Y-axis are expressed as the difference of the web-based uncorrected distance visual acuity assessment outcome minus the ETDRS uncorrected distance visual acuity outcome. The x-axis shows the mean visual acuity in LogMAR of the two assessments, where a more negative value represents a higher visual acuity. The outcome is stratified for a 'better visual acuity' subgroup (uncorrected distance visual acuity ≤0.5 LogMAR) highlighted with a red circle.

**Overall outcome of the web-based refraction assessment.** Web-based and manifest assessments of refractive error are reported stratified for myopia and hyperopia (see Table 2). Detailed outcomes per keratoconus stage are found in the S1 and S2 Tables. Overall, the spherical equivalent of the refractive error measured using the manifest refraction differed from the web-based refraction assessment by 0.09D for myopic subjects (p = 0.675), and -2.06D for hyperopic subjects (p<0.001). Albeit the relatively small and non-significant mean difference in the myopic group, the 95% LoA of the differences of the spherical equivalent extended beyond the *a priori* set non-inferiority limit of 0.5 D in both myopic subjects (95% LoA-3.28–3.47) and hyperopic subjects (95% LoA -5.52–1.39). When transposed into power vectors these differences measure -1.08D for myopic subjects and -0.69D for hyperopic subjects (Table 2, top row).

Fig 3 depicts the Bland-Altman plot of the web-based refraction assessment versus the reference test. The visualization shows a wide distribution of differences between the assessments overall. However, within the emmetropic/myopic refractive error group the agreement between the methods improves in the low refractive error ranges. Furthermore, it can be observed that the web-based refractive assessment yields a more hyperopic outcome in higher myopes (i.e. undercorrected) and a more myopic refractive outcome in low myopics (i.e. overcorrected). When outcomes of the refractive assessment are broken down per keratoconus stage, the more advanced keratoconus cases (AK stage 2 and 3) show an increased difference between the index and reference test (S1 and S2 Tables).

**Table 2. Measured refractive error and visual acuity.**

| Refractive error and visual acuity | Emmetropic and myopic subject | | | | | Hyperopic participants | | | | |
|---|---|---|---|---|---|---|---|---|---|---|
| | Web-based refraction[a] (n = 60) | Manifest Refraction[a] (n = 71) | Difference [b,c] | 95% CI | P-value[d] | Web-based refraction[a] (n = 27) | Manifest refraction[a] (n = 29) | Difference[b,c] | 95% CI | P-value[d] |
| Power vector[e,f] (D) | 2.34 | 2.69 | -1.08 | -1.41 --0.74 | N.A. | 1.35 | 2.02 | -0.69 | -0.92 --0.46 | N.A. |
| | ±1.02 | ±1.74 | | | | ±0.68 | ±1.04 | | | |
| Power vector J0[f] (X) | -0.08 | -0.61 | -1.16 | -1.41 --0.90 | N.A. | -0.08 | -1.27 | -1.47 | -2.00 --0.93 | N.A. |
| | ±0.54 | ±1.17 | | | | ±0.26 | ±1.01 | | | |
| Power vector J45[f] (Y) | 0.05 | 0.02 | -0.72 | -0.91 --0.53 | N.A. | -0.04 | -0.16 | -0.98 | -1.41 --0.54 | N.A. |
| | ±0.62 | ±0.86 | | | | ±0.23 | ±1.00 | | | |
| SEQ (D) | -1.96 | -2.05 | 0.09 | -0.35-0.54 | 0.675 | -1.02 | 1.04 | -2.06 | -2.76 --1.37 | <0.001 |
| | ±1.20 | ±1.76 | | | | ±1.40 | ±0.83 | | | |
| Sphere | -1.53 | -0.87 | -0.76 | -1.23-0.28 | N.A. | -0.59 | 2.68 | -3.28 | -4.22-2.33 | N.A. |
| | ±1.15 | ±1.79 | | | | ±1.39 | ±1.45 | | | |
| Cylinder | -1.00 | -2.44 | 1.58 | 1.06-2.12 | N.A. | -0.93 | -3.14 | 2.21 | 1.39-3.03 | N.A. |
| | ±1.02 | ±1.84 | | | | ±0.91 | ±1.86 | | | |
| Axis | 83 | 92 | -9 | -20-1 | N.A. | 81 | 93 | -53 | -143-37 | N.A. |
| | ±45 | ±45 | | | | ±64 | ±20 | | | |
| CDVA LogMAR | 0.23 | 0.00 | 0.26 | 0.15-0.37 | <0.001 | 0.19 | 0.01 | 0.17 | 0.08-0.27 | 0.001 |
| | ±0.35 | ±0.12 | | | | ±0.27 | ±0.13 | | | |
| CDVA Snellen | 0.72 | 1.03 | 0.38 | -0.51 --0.27 | N.A. | 0.73 | 1.01 | -0.27 | -0.39 --0.16 | N.A. |
| | ±0.37 | ±0.28 | | | | ±0.32 | ±0.26 | | | |

Abbreviations: CDVA: corrected distance visual acuity, CI: confidence interval, D: diopter, N.A.: not assessed, SEQ: spherical equivalent, logMAR: logarithm of the minimum angle of resolution for visual acuity.

[a] Unless otherwise specified, reported as mean ±SD.

[b] Unless otherwise specified, reported as mean difference (web-based minus manifest assessment).

[c] Differences are based on the 54 and 26 cases with both manifest and digital refraction data available, leading to small deviations when subtracting the reported mean data in the table.

[d] Paired-sample Student t test was performed for predefined primary and secondary outcome parameters only.

[e] Spherical and cylindrical power and axes were translated into vectors using Fourier analysis and the difference is calculated as a power vector of the difference between the power vectors.

[f] The difference between power vectors and the vector specific parameters are calculated as a residual vector and is non-linear.

We performed a multivariable Generalized Estimating Equations (GEE) analysis to correct for the inclusion of two eyes of one patient, age, sex, and keratoconus severity (S3 Table). As expected the Amsler-Krumeich stage >1 (B = 1.167, P = 0.027) had a significant effect on the power vector of the manifest refraction, indicating the refractive error increases with keratoconus severity. The web-based refraction did not identify this increase in power vector. Stratification of outcomes for myopia and hyperopia revealed no new insights.

The algorithm of the digital refraction was not always able to correctly determine the participant's refractive error as either myopia (-) or hyperopia (+). In a total of 21 cases (21%) the signation between the index test and the reference test differed, with an average absolute difference in refractive error of -2.38±1.96. Determining the correct signation in hyperopic subject proved challenging: in 20 of 27 (74%) of hyperopic subjects the signation switched between the index and reference test. Strikingly, only one myopic case was incorrectly identified by the online test (98% success). Naturally, this has profound effect on the attained corrected distance visual acuities with the web-based prescription.

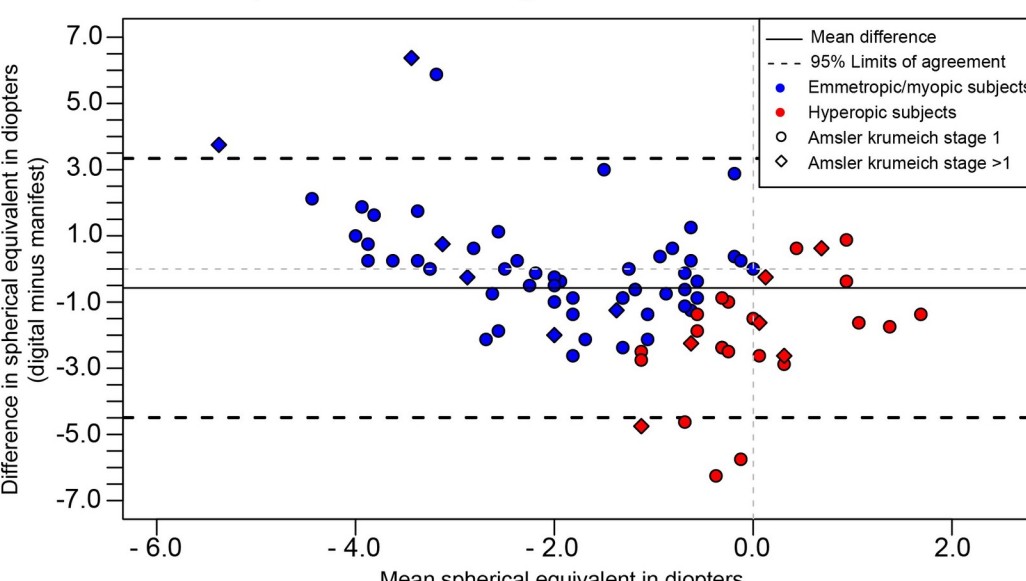

**Fig 3. A Bland-Altman plot displaying the differences in refractive error between the web-based refractive assessment (index test) and the manifest refraction (reference test).** The difference between the reference and index test shown on the Y-axis is expressed as the difference of the web-based refractive assessment outcome compared to the manifest refraction. The x-axis shows the mean spherical equivalent of the two assessments. Myopia and hyperopia were based on the spherical equivalent of the manifest refraction.

**Corrected visual acuity measurements using the web-based prescription.** The overall achieved corrected distance visual acuity was significantly lower with the web-based derived prescription (0.22±0.32 logMAR) versus the reference test (-0.01±0.13 LogMAR, *P* <0.001). For myopic cases the mean difference in CDVA was 0.23 logMAR (95%CI -0.37 to -0.15; Snellen 0.31 95%CI 0.27 to 0.51). For hyperopic cases these outcomes were comparable (-0.22 log-MAR; 95%CI -0.27 to -0.08; Snellen 0.31 95%CI 0.16 to 0.39). This underlines that the variation in refractive outcomes translate to corrected visual acuity outcomes on average 3 lines less read on a visual acuity chart.

In 51 eyes (n = 26 subjects) the CDVA was not assessed with the prescription of the web-based refractive error assessment. In 13 eyes the web-based refraction assessment did not yield an outcome because of the previously mentioned technical errors. The outcomes were particularly missing in cases with higher refractive errors and lower visual acuities and were considered not missing at random. In 19 consecutive assessed patients the CDVA was not assessed due to an incorrect instruction of a member of the research team. We identified no other clinical associations for these 19 patients, and considered these data missing at random.

## Discussion

In this clinical method comparison study we compared a web-based tool for measuring refractive errors with a manifest refraction, the current gold standard, in keratoconus patients. The relevance of delivering remote eye care has been illustrated during the COVID-19 outbreak [4]. With an acute reduction in access to care during this period, the most relevant finding of our studies is that visual acuity can be assessed using a web-based exam, in healthy subjects as well as in individuals with a complex refractive error and an ophthalmic condition [6]. It should be noted that repeated visual acuity assessments always demonstrate variability due to

measurement variation. Cross-sectional studies on repeatability of clinical logMAR wallcharts revealed 95% limits of agreements of +- 0.15 logMAR [21]. When looking at the visualized differences between the web-based test and the reference test, we consider the agreement to be clinically acceptable, particularly in the better visual acuity subgroup (i.e. with visual acuity scores ≤0.5 LogMAR). Notwithstanding, the spread in the outcomes of the web-based refraction assessment exceeded the predefined non-inferiority margins (of <0.5D), and poorly correlated with the gold standard manifest refraction (for myopia 95% LoA: -3.28–3.47, and for hyperopia 95% LoA: -5.52–1.39; ICC: 0.36). As a result, the attained CDVA with the prescription of the web-based tool was significantly lower than the traditional manifest refraction (logMAR 0.23 vs. 0.00, P<0.001). It should be noted that the refractive assessments do not agree equally throughout the range of measurements. The agreement appears to improve in the low myopic refractive error range, suggesting a better performance of the web-based refractive assessment in this range. Furthermore, we observed that the web-based refractive assessment undercorrected higher myopics and overcorrected lower myopics. Both effects might by mitigated by a judicial re-calibration of the algorithm of the web-based test.

The performance of the web-based refractive assessment in this study population can be explained by the design of the algorithm. The algorithm translates the measured visual acuity in a refractive error. Next, the astigmatic refractive error is assessed and the spherical and cylindrical components are determined. The algorithm assumes that the loss of visual acuity is proportional to the increase in refractive error, and that all vision loss is caused by a refractive error. These two assumptions obviously do not necessarily stand in eyes with an ophthalmic condition, such as keratoconus of different stages.

The signation (either + or -) is assessed by a red/green test, and by asking the participant questions on the experience of their visual function (e.g. "do you have problems reading?", "do you recognize faces from afar?" etc.). These questions appear to suffice in a healthy population (98% success) [6], but are not considerable reliable in this population (80% success): keratoconus patients often have problems with both near and far tasks, and the questions have not always been discriminative. This effect is more pronounced in hyperopic refractive errors and could be mitigated by feeding the algorithm more data than was available in the clinical trial, in particular data on any previous prescriptions. This is the case in the commercially available exam: an optometrist assesses any existing prescription and validates the findings that are produced by the algorithm. The here employed clinical trial algorithm functioned completely independent.

Several considerations of the study deserve attention. Firstly, a consideration should be made regarding the missing data. The primary outcome–refractive error- was missing in 13 eyes because of technical errors. The number of technical errors was evidently higher when compared to the healthy study population previously described (13/100 vs. 6/200 eyes) [6]. The data were particularly missing in cases with higher refractive errors and lower visual acuities (i.e. the more severe cases), suggesting these cases are not missing at random. Apparently, the complex refractive errors pose a challenge for the algorithm. This may have impacted the study power. The missing data are considered to have little impact on our conclusions, since we already concluded that the algorithm's performance is poor in this group. Furthermore, no randomization of the test order was performed, which could have impacted our results as subject may become tired during the assessments. However, because of the fixed test order, this should have impacted all subject similarly. The learning or training effect is considered negligible since the three refractive assessment methods are very different and randomized optotypes were used for assessment of the visual acuity. Observer bias cannot be excluded as the observer had access to all test results. However, we consider the risk of bias low as the web-based tool is a self-assessment performed by the patient and the operator has little influence on the

autorefractor outcome. Lastly, the participants were young (25.6 years on average) and presumably digital natives: the uptake of these novel tools in an elderly population warrants a design tailored to their needs and digital aptitude.

The digital refraction fits into the current trend of health care digitalization as health care demand is increasing [22]. Importantly, this is expected to increase because of an aging population, whereas health care budgets are capped, and countries experience a shrinking workforce [23–25]. In addition, telemedicine has the ability to alleviate the urgent challenges our health care systems are currently facing. Several studies have shown the potential of digital eye care tools and the use of digital tools for diagnosis and monitoring of patients [7, 26–29]. In particular, the PEEK vision tool developed by Bastawrous et al., provided health care professionals with accurate and vital information regarding a person's ocular health status, and improved eye health in a rural African community [7]. Notwithstanding, our results show that validation is of upmost importance preceding clinical implementation of these tools and new iterations are needed to further improve the accuracy of the here studied web-based assessment tool.

## Conclusions

The web-based digital eye exam is a promising tool for obtaining visual acuity outcomes, assessed independently and remotely by the patient. The agreement with conventional ETDRS assessments is acceptable, particularly for subjects within a better visual acuity range. The web-based refraction assessment is inferior to a subjective refraction, in this keratoconus population. Contrary to the previously published results with healthy volunteers, the assessment of complex refractive errors posed too big a challenge for the digital algorithm and, consequently, its refraction resulted in a significantly poorer visual performance. These data provide insights in the web-based exam's limitations and will aid in better identification of outliers of the remote assessments. A web-based exam should not be considered a replacement for a comprehensive manual examination by an eye care professional. Notwithstanding, the outcome of the digital eye exam can provide doctors with a quantifiable measurement of visual function which enhances a teleconsultation. The latter is especially important in times or areas with limited access to health care.

## Supporting information

**S1 Checklist. The Standards for Reporting of Diagnostic Accuracy checklist.**
(DOCX)

**S1 Table. The refractive error and visual acuity measured in emmetropic and myopic eyes stratified for keratoconus severity.**
(DOCX)

**S2 Table. The refractive error and visual acuity measured in hyperopic eyes stratified for keratoconus severity.**
(DOCX)

**S3 Table. Multivariate analysis to identify the associations between independent variables and the refractive error outcome for the difference between the web-based refractive assessment and manifest refraction, the web-based refractive assessment and the manifest refraction.**
(DOCX)

## Acknowledgments

The authors are grateful to Nienke Soeters PhD and Veerle Berkhof BSc for their assistance in acquiring the data.

## Author Contributions

**Conceptualization:** Yves F. D. M. Prevoo, Robert P. L. Wisse.

**Data curation:** Marc B. Muijzer.

**Formal analysis:** Marc B. Muijzer, Janneau L. J. Claessens, Daniel A. Godefrooij, Robert P. L. Wisse.

**Funding acquisition:** Yves F. D. M. Prevoo, Robert P. L. Wisse.

**Investigation:** Marc B. Muijzer.

**Methodology:** Marc B. Muijzer, Janneau L. J. Claessens, Daniel A. Godefrooij, Robert P. L. Wisse.

**Project administration:** Marc B. Muijzer.

**Resources:** Francesco Cassano.

**Software:** Francesco Cassano, Yves F. D. M. Prevoo.

**Supervision:** Robert P. L. Wisse.

**Visualization:** Marc B. Muijzer.

**Writing – original draft:** Marc B. Muijzer, Janneau L. J. Claessens.

**Writing – review & editing:** Marc B. Muijzer, Janneau L. J. Claessens, Francesco Cassano, Daniel A. Godefrooij, Yves F. D. M. Prevoo, Robert P. L. Wisse.

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
