## [Decision Letter · Decision Letter 0]

11 Mar 2021

PONE-D-20-36881

The evaluation of a web-based tool for measuring the uncorrected visual acuity and refractive error in keratoconus eyes: a prospective open-label method comparison study

PLOS ONE

Dear Dr. Muijzer,

Thank you for submitting your manuscript to PLOS ONE. After careful consideration, we feel that it has merit but does not fully meet PLOS ONE’s publication criteria as it currently stands. Therefore, we invite you to submit a revised version of the manuscript that addresses the points raised during the review process.

We look forward to receiving your revised manuscript.

Kind regards,

Timo Eppig

Academic Editor

PLOS ONE

Journal Requirements:

4.We note that the grant information you provided in the ‘Funding Information’ and ‘Financial Disclosure’ sections do not match.

5.Thank you for providing the following Funding Statement: 

"This investigator initiated study was sponsored by a grant from Easee BV."

We note that one or more of the authors have an affiliation to the commercial funders of this research study : Easee BV

Reviewers' comments:

Reviewer's Responses to Questions

**Comments to the Author**

1. Is the manuscript technically sound, and do the data support the conclusions?

Reviewer #1: Yes

Reviewer #2: Partly

Reviewer #3: Yes

2. Has the statistical analysis been performed appropriately and rigorously? 

Reviewer #1: Yes

Reviewer #2: I Don't Know

Reviewer #3: Yes

3. Have the authors made all data underlying the findings in their manuscript fully available?

Reviewer #1: Yes

Reviewer #2: Yes

Reviewer #3: Yes

4. Is the manuscript presented in an intelligible fashion and written in standard English?

Reviewer #1: Yes

Reviewer #2: Yes

Reviewer #3: Yes

5. Review Comments to the Author

Reviewer #1: I will focus on methods and reporting.

The abstract is clearly written and balanced.

Overall, the methods appear appropriate, but more references need to be added about the less standard analyses (vectors etc). Some are already there, but a couple more need to be added where appropriate.

the multivariable model is not clear (e.g. how bilaterality modelled, keratoconus severity etc).

I am not convinced by the decision to use complete case analysis. How do the authors know the missingness mechanism? did they test for it? Also a complete case analysis tends to be more problematic that multiple imputation even if data are MNAR, see https://pubmed.ncbi.nlm.nih.gov/28068910/

A major concern is that the power calculation was for the whole trial (which is not clearly reported here, besides the formula). This subgroup analysis is very likely underpowered, In non-inferiority trials this is more crucial, since lower power biases the results towards non-inferiority. The authors either need to demonstrate they have enough power or they need to move away from non-inferiority and present exploratory comparisons, with strong caveats.

Reviewer #2: 1) You are including keratoconus (KC) patients, but you have no clear definition of the methods of such diagnosis?

Did you include overt KC cases only which are diagnosed by signs in the cornea?

Who was responsible to diagnose your cases and with which criteria?

You did not mention anything about clinical signs of your cases. How many of your patients had clinical signs of KC?

2) Your inclusion criteria omit the patients outside the refractive error +4 to -6 diopters of spherical equivalent( SE).

Where those numbers come from?

It suggests that based on some kind of primitive analysis you decided to omit higher SE patients because you had found those patients are not responsive to your algorithms. If not, mention the exact cause of such decision.

Many KC patients' refractions are outside the SE of +4 to -6 diopters and you can not generalize your data to the majority of KC cases. You may excluded many severe cases and this is a source of bias.

In this regard, you can not mention the general term of KC in your topic and you have to say 'KC cases with certain refractive error'.

3)In your exclusion criteria, you only mentioned diabetes as a cause of exclusion. What about other retinal or optic nerve problem?

Did you include glaucoma patients , other retinal disorders like RP case and macular hole, cataract, PCO ,media opacity,  etc ? As we know all those cases may have some problems in contrast sensitivity and visual acuity.

4) You mentioned 24 patients have a history of previous ocular surgery. Mention the type of surgeries for further clarification.

5) In the last paragraph of your results( line 275), you mentioned that you could not assess the CDVA of more than half of your patients with your web system. In 13 of eyes evaluation did not have an outcome and 19 eyes had a problem because of bad instruction. These cases are a large proportion of your sample size and you cannot overlook this large numbers by saying " missing at random" !!!.

You have to separately speak about this data by clearly defining their refractive error, severity, etc in a separate table.

Reviewer #3: The authors tested the possibility of a web-based too that measures visual acuity to translate the visual acuity results into refractive correction in patients with various amounts of presbyopia. As the authors present original research that is of interest for the field, especially with the background of field testing of vision, the methods need to be detailed and the data needs some more work regarding the interpretation.

Abstract: The authors did not mention that they assessed variability but present data on it – how was it assessed?

Line 66: I am not sure about that the content of the statement the authors want to make. For sure, an autorefractor measures refractive errors. But out of that data, it is possible to make an assumption of the uncorrected visual acuity. Also, there might be autorefractors on the market that allow to measure visual acuity (either corrected as well as uncorrected).

Line 91: could patient acquisition has been biased by the use of already known patients of the clinic? This could also influence the (subjective) refractive measurements, as the operator might not have been blind to the medical history of each patient. The authors should also add some details on the question if operator was blind for the measurements of the refractive errors and if the same operator did all the measurements?

Line 102: why was the testing not randomized between subjects? What a about a possible learning effect in visual acuity testing?

Line 120: I guess that correction was inserted in a trial frame and trial lenses were used? Was refractive correction standardized for the same back vertex distance.

Line 129: if subjects were normally wearing contact lenses, how long was the wash-out time before participants could participate in the study?

Line 156: due to the mirrorsymmetrie of both eyes, only one should be analyzed. Is the use of the multivariable generalized estimates equation a method to reduce potential bias?

Line 164: this statement should not be parts of the Material / Methods section

Line 174: the web-based tool also measured myopia in several patients with hyperopic refractive errors. Is this a typical behavior also in subjects without any ocular diseases? What would be the implication for a screening?

Line 183: is this a typical of measurements that are missing?

Line 190: what was the ICC for the autorefractor vs. web based measurement and autorefractor vs. manifest refraction?

Line 202: the definition of the low and high visual acuity group should be done in the methods section.

Line 225: what was the significance value?

Line 233: is this because the level of keratoconus is smaller in this group?

6. PLOS authors have the option to publish the peer review history of their article (what does this mean?). If published, this will include your full peer review and any attached files.

Reviewer #1: No

Reviewer #2: No

Reviewer #3: No

---

## [Author Response · Author response to Decision Letter 0]

17 May 2021

Rebuttal manuscript 

Manuscript ID: PONE-D-20-36881

Subject: rebuttal for manuscript “The evaluation of a web-based tool for measuring the uncorrected visual acuity and refractive error in keratoconus eyes: a prospective open-label method comparison study”

Reviewers' comments:

Reviewer #1: 

I will focus on methods and reporting. The abstract is clearly written and balanced.

1. Overall, the methods appear appropriate, but more references need to be added about the less standard analyses (vectors etc). Some are already there, but a couple more need to be added where appropriate.

Reply: We have included additional relevant literature on the subject of power vectors and changed were the literature is cited to be more appropriate with the information provided. 

2. the multivariable model is not clear (e.g. how bilaterality modelled, keratoconus severity etc).

Reply: In the study we corrected for bilaterality (two eyes of the same subject) using a generalized linear model (i.e. generalized estimating equation). This model tests the independent association for each variable correcting for two eyes of the same patient with an unknown association. The model used is comparable to a linear mixed-model regression and frequently used for ophthalmologic studies. The keratoconus severity was determined using the Amsler-Krumeich staging as a covariate. We have clarified both in the manuscript (see line: 170-171) 

3. I am not convinced by the decision to use complete case analysis. How do the authors know the missingness mechanism? did they test for it? Also a complete case analysis tends to be more problematic that multiple imputation even if data are MNAR, see https://pubmed.ncbi.nlm.nih.gov/28068910/

 Reply: 

The author addresses an important point. We are aware of the missingness mechanism as we have tested for association of missing cases with other parameters. In 13 eyes the outcome was missing because of technical errors in the web-based tool and associated with a higher refractive error. High refractive errors appeared more challenging for the web-based tool, and these are more prevalent in severe cases of keratoconus These missing data where therefore considered MNAR.

In the remaining 19 cases the missing outcomes were associated with a timeframe, during which one member of the study team did not take the CDVA with the online refraction in a trial frame. This behaviour was non-compliant with the study procedures,, and the outcomes were not associated with relevant patient characteristics. They were therefore considered MAR. We agree with the reviewer that a complete case analysis can be problematic and tried to impute these missing data. Unfortunately - because of the heterogeneity between cases and the extent of missing data - imputing the data proved impossible. 

Notwithstanding, we do not expect that the conclusions we draw would materially differ between imputing or complete case analysis: we conclude that the complex refractive errors of keratoconus patients appeared challenging for the algorithms of the web-based tool and do not intent to make claims on the validity of the web-based assessment of refractive error in this group of patients. Imputing data will not alter this conclusion. 

4. A major concern is that the power calculation was for the whole trial (which is not clearly reported here, besides the formula). This subgroup analysis is very likely underpowered, In non-inferiority trials this is more crucial, since lower power biases the results towards non-inferiority. The authors either need to demonstrate they have enough power or they need to move away from non-inferiority and present exploratory comparisons, with strong caveats.

Reply: We thank the reviewer for this comment. The current study was indeed part of an overarching prospective trial with both healthy and keratoconus patients, as can be seen on clinicaltrials.gov/NCT03313921. The results of the healthy peers are referred to extensively throughout this paper (Wisse et al. JMIR 2019), and the study was powered on this healthy subgroup. We expected upfront that this keratoconus group with complex refractive errors could provide a big challenge for the algorithms. Advancing methodological insights even made the non-inferiority margins stricter in this current manuscript (when compared to the previous publication): we consider 95% levels of agreements rather than mean differences. 

We agree that the current method section is confusing and does deserve more explanation regarding abovementioned. We have rewritten the method section to better introduce the two groups of the MORE trial, removed the power calculation formula pertaining to the healthy subgroup, and included a post-hoc power calculation for the keratoconus group to prove that we have sufficient power to assess the non-inferiority limit of ± 0.5.

Reviewer #2: 

1. You are including keratoconus (KC) patients, but you have no clear definition of the methods of such diagnosis? Did you include overt KC cases only which are diagnosed by signs in the cornea? Who was responsible to diagnose your cases and with which criteria? You did not mention anything about clinical signs of your cases. How many of your patients had clinical signs of KC?

Reply: The diagnosis of keratoconus was established by an ophthalmologist specialized in corneal diseases based on clinical signs and Scheimpflug corneal tomography. As our center is a tertiary expertise & referral center, all patients had clinical signs of keratoconus. We have clarified this section in the manuscript, and added that the involved corneal specialist graded all cases accordingly. 

2. Your inclusion criteria omit the patients outside the refractive error +4 to -6 diopters of spherical equivalent( SE). Where those numbers come from? It suggests that based on some kind of primitive analysis you decided to omit higher SE patients because you had found those patients are not responsive to your algorithms. If not, mention the exact cause of such decision.

Reply: These inclusion criteria do not pertain directly to the keratoconus population, but to the design of the overall trial, including healthy individuals. In the revised version of the manuscript we’ve made it much clearer in the method section that these data pertain to a keratoconus subgroup of the larger MORE trial. The outcomes of the web-based tool are referred to extensively in the manuscript (Wisse et al. JMIR 2019). When we designed the trial, we aimed to assess the validity of web-based refraction for people with relatively normal refraction errors. Owing to our status as an academic expert center for keratoconus care, we felt obliged to study the validity in this group with complex refractive errors. The +4 / -6D ranges are dictated by the algorithm, not by the clinical population. This is now better explained in the methods section. 

As such, there was no primitive analysis, and the inclusion of keratoconus patients followed chronologically in time after the healthy subjects. But we understand the reviewers remarkt, and are confident that a better structured methods section now prevents readers from reaching the same conclusion. 

3. Many KC patients' refractions are outside the SE of +4 to -6 diopters and you can not generalize your data to the majority of KC cases. You may excluded many severe cases and this is a source of bias. In this regard, you can not mention the general term of KC in your topic and you have to say 'KC cases with certain refractive error'.

Reply: We understand the important suggestion of the reviewer regarding the generalizability of our results for a complete KC population. But we do not agree that we should say ‘KC cases with certain refractive error’. The study population is a representative sample of the patient population in our tertiary expert center for keratoconus. The majority of our population had an Amsler Krumeich classification of either 1 or 2 and their refractive error fell within the test limits of the algorithm. More importantly, the complex refractive errors seen in keratoconus patients appeared challenging for the web-based tool: high refractive errors led to missing data, the signation (+ vs -) was not robust (79% correct), and the correlation between the web-based assessment and in-hospital manifest refraction was rather poor (ICC 0.32). Our conclusion is that a web-based assessment of refraction in keratoconus patients should not be done. There is no reason to assume that this advice would be different for keratoconus cases with even higher refractive errors or Amsler-Krumeich grading. The visual acuity assessments proved more robust, and this was retained in the conclusion of the manuscript.

4. In your exclusion criteria, you only mentioned diabetes as a cause of exclusion. What about other retinal or optic nerve problem? Did you include glaucoma patients , other retinal disorders like RP case and macular hole, cataract, PCO ,media opacity, etc ? As we know all those cases may have some problems in contrast sensitivity and visual acuity.

Reply: The inclusion criteria for this keratoconus study were commensurate to the larger MORE study in healthy subjects, where any ophthalmic condition or previous surgery led to study exclusion. Apparently, for this keratoconus subgroup report, we failed to explicitly state that the subjects were devoid of other pathologies than keratoconus. None of the subjects had co-existing visual acuity limiting conditions. This method section was extended regarding this aspect.

5. You mentioned 24 patients have a history of previous ocular surgery. Mention the type of surgeries for further clarification.

Reply: We thank the reviewer for noticing this textual error. In the text the procedure performed was specified (in all cases corneal crosslinking with >6m follow-up). 

6. In the last paragraph of your results( line 275), you mentioned that you could not assess the CDVA of more than half of your patients with your web system. In 13 of eyes evaluation did not have an outcome and 19 eyes had a problem because of bad instruction. These cases are a large proportion of your sample size and you cannot overlook this large numbers by saying " missing at random" !!!. You have to separately speak about this data by clearly defining their refractive error, severity, etc in a separate table.

Reply: We agree our formulation of this paragraph may give a false impression we assume the technical errors to be missing at random. In 13 eyes the outcome was missing because of technical errors in the web-based tool and associated with a higher refractive error. High refractive errors appeared more challenging for the web-based tool, and these are more prevalent in severe cases of keratoconus. These missing data where therefore considered not missing at random.

In the remaining 19 cases the missing outcomes were associated with a timeframe, during which one member of the study team did not take the CDVA with the online refraction in a trial frame. This behaviour was non-compliant with the study procedures, and the outcomes were not associated with relevant patient characteristics. They were therefore considered missing at random. . We have clarified this specific paragraph. 

Reviewer #3: The authors tested the possibility of a web-based too that measures visual acuity to translate the visual acuity results into refractive correction in patients with various amounts of presbyopia. As the authors present original research that is of interest for the field, especially with the background of field testing of vision, the methods need to be detailed and the data needs some more work regarding the interpretation.

1. Abstract: The authors did not mention that they assessed variability but present data on it – how was it assessed?

Reply: The term “variability” addresses the 95%LoA, meaning the “variability of the differences” in the Bland-Altman analysis. We have clarified this section of the abstract. 

2. Line 66: I am not sure about that the content of the statement the authors want to make. For sure, an autorefractor measures refractive errors. But out of that data, it is possible to make an assumption of the uncorrected visual acuity. Also, there might be autorefractors on the market that allow to measure visual acuity (either corrected as well as uncorrected).

Reply: Previous research from our study group indicated that the autorefractor measurements in irregular keratoconus corneas are not reliable (Soeters N, Muijzer MB, Molenaar J, Godefrooij DA, Wisse RPL. Autorefraction Versus Manifest Refraction in Patients With Keratoconus. J Refract Surg. 2018;34(1):30–4). Subsequently it is not possible to calculate or derive an exact visual acuity from these measurements (other than a rough estimation). To our knowledge there are no devices on the market that measure exact values of visual acuity as a visual acuity measurement is a psychophysical test. The here presented web-based tool is a self-assessment and does not require specialized equipment or medically trained personal. In this revision we made it more clear that this report is a subgroup of the larger MORE-trial in healthy subjects, which is referred to extensively. In these healthy subjects, auto-refraction measurements are reliable estimates, and we considered that a valid additional test to put the web-based outcomes in perspective. The study design for this keratoconus group is exactly the same, but upfront we knew the autorefractor measurements would be of little value. For methodological clarity and completeness we choose to also report these outcomes.

3. Line 91: could patient acquisition has been biased by the use of already known patients of the clinic? This could also influence the (subjective) refractive measurements, as the operator might not have been blind to the medical history of each patient. The authors should also add some details on the question if operator was blind for the measurements of the refractive errors and if the same operator did all the measurements?

Reply: In practice, the corneal specialist and specialized optometrists invited their patients for the MORE trial, and involved the executive researcher subsequently. These clinicians were aware of the disease status/refractive error/clinical data and an inclusion bias can therefore not be excluded. Notwithstanding, the clinicians were not involved in the collection of research data/web-based measurements. Nor the operator, nor the patients could be blinded in this open-label study., Notwithstanding, we consider the effect of bias negligible, because the three assessments of refractive error differ in nature. The operator has virtually no influence on the auto-refractor measurements and the web-based tool is a self-assessment that reports an objective outcome which was recorded without interpretation. We included this remark in the manuscript. 

4. Line 102: why was the testing not randomized between subjects? What a about a possible learning effect in visual acuity testing?

Reply: We considered taken the 3 tests in a random order. Rather we choose to change the optotypes randomly for each visual acuity measurement. Therefore,we considered the learning effect negligible Now, the order of test was the same in all patients so that any learning impact would be similar for all patients. 

5. Line 120: I guess that correction was inserted in a trial frame and trial lenses were used? Was refractive correction standardized for the same back vertex distance.

Reply: Yes, the final measurement was a CDVA with the prescription derived from the web-based assessment. We thank the reviewer for this detailed comment. We haven’t standardized the refractive correction for the vertex distance. In practice, the trial frame was fitted by a trained optometrist with extensive knowledge on this particular matter, and no irregularities were recorded.. So, technically, in some subjects with large refractive errors the vertex distance could have influenced their visual acuity outcome, however, we do not consider this effect clinically relevant, nor will it impact our conclusions. We have included this consideration in the discussion section.

6. Line 129: if subjects were normally wearing contact lenses, how long was the wash-out time before participants could participate in the study?

Reply: All patient included in the study have had an 4 week washout period commensurate to our clinical protocol. 

7. Line 156: due to the mirror symmetrie of both eyes, only one should be analyzed. Is the use of the multivariable generalized estimates equation a method to reduce potential bias?

Reply: The reviewed makes an important remark on including both eyes of one patients for statistical analysis. We corrected for bilaterality using a generalized estimating equation analysis, which distributes confounding factors at a within-person or within-group level. This has become common practice in ophthalmic studies and increases the power of a study, though not by a factor 2. In keratoconus studies in particular, where the disease can be very asymmetric, there is good reason to include both eyes: here the within-person variation would be lost if only one eye was considered. We report the outcomes of this analysis extensively in supplementary table 3.

8. Line 164: this statement should not be parts of the Material / Methods section

 Reply: we agree with the reviewer and removed this sentence in the methods section. 

9. Line 174: the web-based tool also measured myopia in several patients with hyperopic refractive errors. Is this a typical behavior also in subjects without any ocular diseases? What would be the implication for a screening?

Reply: This is not typical behaviour in these patients, but a technical limitation of the presented algorithm. The refractive error is measured a an absolute value (without plus/minus indication) and based on a questionnaire and duochrome test the signation is assigned. Keratoconus patient have a range of symptoms which can be seen in both myopic and hyperopic patient. This method of assigning the signation proved especially challenging for hyperopic patients who encounter visual acuity difficulties for both distance and near tasks. We have addressed this in the discussion section. For screening purposes this does not have implications, as one would mostly be interested in detecting visual impairment (i.e. low visual acuity). 

10. Line 183: is this a typical of measurements that are missing?

Reply: We don’t completely understand the question of the reviewer. In 13 eyes the outcome was missing because of technical errors in the web-based tool and associated with a higher refractive error. High refractive errors appeared more challenging for the web-based tool, and these are more prevalent in severe cases of keratoconus These missing data where therefore considered not missing at random. This is now addressed in the discussion section.

11. Line 190: what was the ICC for the autorefractor vs. web based measurement and autorefractor vs. manifest refraction?

Reply: As mentioned above, the study design for this keratoconus group is exactly the same as for the healthy individuals, and upfront we knew the autorefractor measurements would be of little value. We deliberately decided to report the overall ICC, and don’t consider these other outcomes of interest for our study. First, because we do not study the autorefractor, and second because previous research showed that autorefractor measurements in keratoconus eyes are inaccurate (Soeters N, Muijzer MB, Molenaar J, Godefrooij DA, Wisse RPL. Autorefraction Versus Manifest Refraction in Patients With Keratoconus. J Refract Surg. 2018;34(1):30–4). 

12. Line 202: the definition of the low and high visual acuity group should be done in the methods section.

 Reply: We have clarified this in the methods section. 

13. Line 225: what was the significance value?

 Reply: The significance values have been included. 

14. Line 233: is this because the level of keratoconus is smaller in this group?

Reply: The agreement between the methods improves in the low refractive error ranges. Uncorrected visual acuity assessments will be more accurate in this range, making it easier for the web-based algorithm to determine the refractive error. To answer the question of the reviewer, we state that in general, keratoconus severity is lower in the low refractive error ranges. 

Funding statement: This investigator initiated study was sponsored by Easee BV. The funder provided support in the form of salaries for author [FC], but did not have any additional role in the data analysis, decision to publish, or preparation of the manuscript. The specific roles of these authors are articulated in the ‘author contributions’ section.

Author contributions: The following authors were involved in acquiring funding (RW), study design (RW, DG, YP), developing the algorithm (YP, FC), data collection (MM, FC), data analysis and interpretation (MM, JC, DG, RW), preparation of the manuscript and critical revision (MM, JC, DG, RW), decision to publish (MM, JC, RW). 

Competing interest: MM is a consultant for Easee BV, FC is an employee of Easee BV, YP is the CEO/founder and shareholder of Easee BV, RW is a consultant and shareholder of Easee BV. The competing interest of the authors do not alter our adherence to PLOS ONE policies on sharing data and materials.

---

## [Decision Letter · Decision Letter 1]

10 Jun 2021

PONE-D-20-36881R1

The evaluation of a web-based tool for measuring the uncorrected visual acuity and refractive error in keratoconus eyes: a prospective open-label method comparison study

PLOS ONE

Dear Dr. Muijzer,

Thank you for submitting your manuscript to PLOS ONE. After careful consideration, we feel that it has merit but does not fully meet PLOS ONE’s publication criteria as it currently stands. Therefore, we invite you to submit a revised version of the manuscript that addresses the points raised during the review process.

There are some additional minor comments from Reviewer #3 which shall be considered. 

We look forward to receiving your revised manuscript.

Kind regards,

Timo Eppig

Academic Editor

PLOS ONE

Journal Requirements:

Reviewers' comments:

Reviewer's Responses to Questions

**Comments to the Author**

1. If the authors have adequately addressed your comments raised in a previous round of review and you feel that this manuscript is now acceptable for publication, you may indicate that here to bypass the “Comments to the Author” section, enter your conflict of interest statement in the “Confidential to Editor” section, and submit your "Accept" recommendation.

Reviewer #2: All comments have been addressed

Reviewer #3: All comments have been addressed

6. Review Comments to the Author

Reviewer #3: Line 36 306: visual acuity measurement is not objective. What is the typical intra-individual variation of the web-based test in this patients? (0.1logMAR as with standard acuity test)? As you discuss this in line 306 for VA measurements in subjects with no ocular diseases, do you have data on subjects with keratoconus?

Line 38 and following: statement biased by the sponsor? I doubt that algorithm is trainable with this data.

General: when you only measure VA, how easy is it to diagnose for keratoconus, especially in early stages? Its not, or? In case of “telemedicine” any drop in VA is most likely related to a refractive error and no diagnosis can be made w/o corneal topography. Is the last section of the introduction still valid having that in mind?

Line 284: please change the order and put the limitation on the number of available measurements first. This makes it easier to the reader to classify the results. Please also put the number of cases in brackets for available measurements.

Line 318, please add: as such with keratoconus of different stages.

Line 361: is this statement true for patients without diseases? Please be specific

Line 364: results did not only resulted in poor visual performance (as in terms of VA), but also in a correction that would not be acceptable (in terms of diopters).

Line 368: Again: VA is subjective, not objective.

---

## [Author Response · Author response to Decision Letter 1]

16 Jul 2021

Manuscript ID: PONE-D-20-36881R1

Subject: rebuttal for manuscript “The evaluation of a web-based tool for measuring the uncorrected visual acuity and refractive error in keratoconus eyes: a prospective open-label method comparison study”

Reviewers' comments:

Reviewer #3: All comments have been addressed

Reviewer #2: All comments have been addressed

Reviewer #1: 

Line 36 306: visual acuity measurement is not objective. What is the typical intra-individual variation of the web-based test in this patients? (0.1logMAR as with standard acuity test)? As you discuss this in line 306 for VA measurements in subjects with no ocular diseases, do you have data on subjects with keratoconus?

Reply: We believe that a visual acuity assessment will enhance teleconsultations, but agree with the reviewer a visual acuity test is not an objective measurement and thus removed this word from the abstract and manuscript and replaced it with “quantifiable” (as it provides the health care professional with an exact value). 

We did not investigate the test-retest performance of the web-based assessment, so the intra-individual variation between measurements is unknown. The reported variation between visual acuity measurements that we refer to in the manuscript is derived from a study performed in an eye clinic in patient with various ocular diseases, however, no data specifically for keratoconus patients is available. 

Line 38 and following: statement biased by the sponsor? I doubt that algorithm is trainable with this data.

Reply: We thank the author for this comment. It is indeed true that the algorithm’s calculations cannot with this inadequate data, so we agree that this statement needs some nuance. The study has provided us with valuable insights in the limitations of this web-based tool. We will use this knowledge to increase overall safety of the web-based tool by better identification of outlier cases. We have adjusted this sentence in the Abstract Conclusion and Manuscript conclusion accordingly. 

General: when you only measure VA, how easy is it to diagnose for keratoconus, especially in early stages? Its not, or? In case of “telemedicine” any drop in VA is most likely related to a refractive error and no diagnosis can be made w/o corneal topography. Is the last section of the introduction still valid having that in mind?

Reply: We agree that using visual acuity only keratoconus cannot be diagnosed and to our knowledge we do not claim or suggest this in the manuscript. The aim and focus of this study was to evaluate the performance of the algorithm in complex refraction. Keratoconus patients often/always have a complex refraction (e.g., irregular astigmatism) and were therefore a suitable study population for our research question. In the general population, complex refractive errors are also present with or without an underlying eye condition and this study provided insight in the limitations and accuracy (of in this case inaccuracy) of the algorithm in these cases. 

Line 284: please change the order and put the limitation on the number of available measurements first. This makes it easier to the reader to classify the results. Please also put the number of cases in brackets for available measurements.

Reply: We thank the reviewer for this comment. We have changed the order of the discussed limitations as suggested by the reviewer. We have added the “13/100” in between brackets to clarify the ratio between missing data and available measurements. 

Line 318, please add: as such with keratoconus of different stages.

 Reply: We’ve revised the sentence and included this suggestion. 

Line 361: is this statement true for patients without diseases? Please be specific

Reply: We revised the sentence to make it clear we are referring to our study outcomes. In addition, to answer the question; it is indeed true that visual acuity can be more reliably assessed in better visual acuity ranges regardless of ocular conditions. 

Line 364: results did not only resulted in poor visual performance (as in terms of VA), but also in a correction that would not be acceptable (in terms of diopters).

Reply: For clarification, we have included in the conclusion that the web-based refractive assessment was found to be inferior to the manifest assessment. However, we deem a prescription unacceptable based on visual performance rather than the exact dioptric values as these can differ without directly affecting visual performance. 

Line 368: Again: VA is subjective, not objective.

Reply: see comment 1. 

Other changes:

We replaced “higher visual acuity” subgroup with “better visual acuity” in the legend and footnote of figure 2, so it is now similar to the manuscript.

---

## [Decision Letter · Decision Letter 2]

2 Aug 2021

The evaluation of a web-based tool for measuring the uncorrected visual acuity and refractive error in keratoconus eyes: a method comparison study

PONE-D-20-36881R2

Dear Dr. Muijzer,

We’re pleased to inform you that your manuscript has been judged scientifically suitable for publication and will be formally accepted for publication once it meets all outstanding technical requirements.

Kind regards,

Timo Eppig

Academic Editor

PLOS ONE

Additional Editor Comments (optional):

Reviewers' comments:

Reviewer's Responses to Questions

**Comments to the Author**

1. If the authors have adequately addressed your comments raised in a previous round of review and you feel that this manuscript is now acceptable for publication, you may indicate that here to bypass the “Comments to the Author” section, enter your conflict of interest statement in the “Confidential to Editor” section, and submit your "Accept" recommendation.

Reviewer #2: All comments have been addressed

Reviewer #3: All comments have been addressed

2. Is the manuscript technically sound, and do the data support the conclusions?

Reviewer #2: Yes

Reviewer #3: Yes

3. Has the statistical analysis been performed appropriately and rigorously? 

Reviewer #2: Yes

Reviewer #3: Yes

4. Have the authors made all data underlying the findings in their manuscript fully available?

Reviewer #2: Yes

Reviewer #3: Yes

5. Is the manuscript presented in an intelligible fashion and written in standard English?

Reviewer #2: Yes

Reviewer #3: Yes

6. Review Comments to the Author

Reviewer #2: The manuscript is well written. Although it would be better to do further research in this area with larger sample size,I personally agree that it should be published because of Novelty.

Reviewer #3: (No Response)

7. PLOS authors have the option to publish the peer review history of their article (what does this mean?). If published, this will include your full peer review and any attached files.

Reviewer #2: No

Reviewer #3: No

---

## [Editor Report · Acceptance letter]

10 Aug 2021

PONE-D-20-36881R2 

The evaluation of a web-based tool for measuring the uncorrected visual acuity and refractive error in keratoconus eyes: a method comparison study 

Dear Dr. Muijzer:

I'm pleased to inform you that your manuscript has been deemed suitable for publication in PLOS ONE. Congratulations! Your manuscript is now with our production department. 

Kind regards, 

on behalf of

Dr. Timo Eppig 

Academic Editor

PLOS ONE